# Relations between Video Game Engagement and Social Development in Children: The Mediating Role of Executive Function and Age-Related Moderation

**DOI:** 10.3390/bs13100833

**Published:** 2023-10-11

**Authors:** Ke Xu, Shuliang Geng, Donghui Dou, Xiaocen Liu

**Affiliations:** 1College of Preschool Education, Capital Normal University, Beijing 100048, China; 18336930031@163.com (K.X.); 2223102004@cnu.edu.cn (S.G.); 2School of Sociology and Psychology, Central University of Finance and Economics, Beijing 100081, China; psychaos@126.com

**Keywords:** child, electronic game, social behavior, executive function, age

## Abstract

The global proliferation of video games, particularly among children, has led to growing concerns about the potential impact on children’s social development. Executive function is a cognitive ability that plays a crucial role in children’s social development, but a child’s age constrains its development. To examine the association between video game engagement and children’s social development while considering the mediating role of executive function and the moderating role of age, a questionnaire was distributed to a sample of 431 parents. The results revealed a negative relation between video game engagement and social development in children, with executive function found to mediate this relation fully. Additionally, the negative association between video game engagement and executive function became more pronounced as children grew older. In light of these findings, it is advisable to adopt proactive strategies to limit excessive video game use, consider the developmental characteristics of children at different ages, and prioritize the promotion of executive function to facilitate social development among children.

## 1. Introduction

Video games have rapidly emerged as a prevalent facet of children’s daily experiences globally. Their appeal to children can be attributed to features such as engaging content, immediate feedback, user interactivity, broad media accessibility, and applicability across a diverse age spectrum [1]. A study by Liu et al., encompassing urban and rural populations in China, revealed that 96.1% of children had encountered video games by the age of six. Alarmingly, within this cohort, 36.3% of children engaged in sessions exceeding an hour, with some sessions surpassing eight continuous hours [2]. Alanko noted that a mere 10% of U.S. children aged over two years remain unexposed to video games. Furthermore, youth between 8 and 17 years reportedly dedicate an average of 1.5 to 2 h daily to such digital pursuits [3].

The term “video games” describes games played on audiovisual devices that follow a particular narrative [4]. In English, “video game” and “electronic game” are often used interchangeably, although the latter encompasses all interactive games played on electronic devices. Given the ubiquity of video games within the realm of electronic games, “video game” has become the predominant term for such digital entertainment in English [5]. Previous research primarily relied on objective metrics, such as duration and frequency, to quantify individuals’ interactions with video games [6]. To provide a more nuanced understanding of children’s interactions with video games, the metric of “video game engagement” has been proposed. This concept delineates the intensity of individuals’ motivation and cognitive-emotional stimulation during gameplay [7]. Qu et al. expanded on this by categorizing “video game engagement” into four dimensions: interest in the activity, focus during play, challenges in discontinuation, and social disengagement [8].

The Ecological Techno-Microsystem Theory articulates that diverse media environments, encompassing the use of communication, information, and recreation technologies (i.e., the techno-subsystem), act as an immediate system, profoundly impacting multiple facets of children’s development, including their social, emotional, physical, and cognitive growth [9]. As a form of new media highly favored by children, video games have long been debated regarding their impact on child development. Some argue that video games may positively affect individuals’ cognitive, motivational, emotional, and social development [10,11]. However, evidence suggests that video games, particularly those marketed to young children, can adversely affect academic functioning and child development [12]. In their seminal work, Raventós and Belli shed light on the long-standing biases shaping the perception of video games. They pointed out the historical tendency to focus on the purported negative impact of gaming on youth and society [13]. The authors advocate for a more nuanced, evidence-based understanding of gaming’s societal and individual effects. Consequently, elucidating the impact of video gaming on child development is vitally important, particularly in a post-pandemic society grappling with the long-term consequences of substituting technology for face-to-face interactions.

Notably, the COVID-19 pandemic has fueled an uptick in video game engagement. Supporting this observation, Chen et al. observed that pandemic-induced school closures and limited outdoor activities led to heightened video game exposure among children [14]. Similarly, during the peak of these closures, Donati et al. documented a notable surge in children’s video game usage [15]. Research by Zhu et al. revealed that in a sample of 2863 children and adolescents, 18% engaged in gaming for over three hours per day [16]. Shoshani et al. corroborated this pattern, finding increased gaming among 1537 Israeli youth during the pandemic, a notably pronounced trend for children with ADHD [17]. Werling et al. assessed shifts in media usage through parental reports at three time points (before the blockade, during the blockade, and within 1–2 months afterward), highlighting a 46% increase in children’s media consumption, including video games, during lockdown periods [18]. Given the surge in children’s video game engagement during the pandemic, this study aims to investigate its associations with cognitive and social development, especially in a societal context increasingly shaped by technology-driven youth behavior.

### 1.1. Video Game Engagement and Children’s Social Development

“Social development” refers to the unique psychological characteristics of an individual that are formed on a physiological basis as they participate in social life and interactions. This term encompasses adherence to societal norms, emotional stability, social awareness, familial bonds, and peer relationships [19]. Positive social development during childhood is correlated with favorable social adjustment, effective interpersonal interactions, academic success, and cognitive capacity in subsequent stages of life [20,21]. Thus, understanding the determinants influencing children’s social development is imperative. Extensive research underscores video games’ substantial influence on children’s social development [22,23].

The impact of video games on children’s social development is a topic of debate. Several studies have suggested that video games foster positive outcomes in children’s social growth. For example, Yilmaz and Griffiths posited that the diverse interactions facilitated by pro-social video games bolster the development of children’s peer relationships [24]. Concurrently, other scholars have emphasized the role of video games in enhancing social interaction and collaborative efforts [25]. Many of these games feature multiplayer functionality, fostering improved communication and cooperative skills among children [26]. In consonance with this view, Raventós and Belli indicated that gaming environments enable players to forge friendships and establish shared interests, catalyzing the genesis of novel social bonds [13]. Furthermore, certain video games are geared towards bolstering problem-solving abilities, analytical reasoning, and innovative cognition, all crucial for comprehensive social development [27].

Nonetheless, academic voices are concerned about the ramifications of heightened video game engagement on children’s social development [6]. Davis et al. asserted that children deeply immersed in video games often exhibit increased aggressive behaviors and notable social inadequacies [28]. The General Learning Model postulates that individuals deeply engrossed in video games might develop an addiction-like relationship with the medium, consequently impeding their social development [29]. This model suggests that children heavily involved with video games may consistently internalize the game’s content and mechanics, leading to cognitive and emotional disturbances, including hostility and indifference [30,31]. Further, Gentile et al., through a biennial longitudinal study, discerned that augmented social challenges and impulsivity were byproducts of children’s heightened video game engagement [32]. Thus, excessive video game engagement may negatively correlate with children’s social development.

Undoubtedly, the pandemic’s influence on individual social development warrants scholarly attention. To contain COVID-19, numerous governmental interventions, such as self-quarantine and social distancing, have been put into place [33,34]. However, the pandemic’s far-reaching implications for all demographics, including children, cannot be overlooked. Countries like Germany, China, and Bangladesh have reported increased psychological stress related to health concerns and economic instability [35,36,37], which inevitably has a downstream impact on children’s social development [38]. Additionally, the overwhelming influx of pandemic-related news has heightened anxiety, particularly among children [39]. Nevertheless, contrasting views exist; Allen et al. found minimal changes in children’s emotional well-being pre- and post-pandemic when assessed using Rumble’s Quest [40].

In this complex landscape, infants and young children born or raised during COVID-19 are of specific concern. Lockdown measures have curtailed their opportunities for conventional social interaction and development [41]. In this restricted environment, their engagement with video games becomes a salient point of inquiry. While some research suggests that video games can offer virtual interactions that teach pro-social behaviors like sharing and cooperation [42], others argue that excessive gaming could discourage real-world social engagement, thus impeding social development [43]. In light of this intricate and multifaceted context, it could be hypothesized that the surge in video game engagement, especially induced by the pandemic, may adversely affect children’s social development.

### 1.2. The Mediating Role of Executive Function

Executive function denotes a higher-order cognitive capacity, facilitating the cohesive and analytical regulation of an individual’s cognitive processes and behaviors. This capacity encompasses vital facets such as inhibition, working memory, and cognitive flexibility [44]. These components collectively constitute pivotal constituents within the scaffolding of intricate proficiencies and aptitudes in human development [45,46]. Inhibition is a sophisticated cognitive process that empowers individuals to quell predominant reactions and counteract extraneous interference [47]. Working memory pertains to an individual’s proficiency in continuously retaining, manipulating, and modifying the contents held in short-term memory [48]. Cognitive flexibility is the disposition to engage in creative ideation, adopt diverse perspectives, and promptly and adaptively respond to altered contexts [44]. The underpinning of executive function is closely intertwined with the prefrontal cortical region of the cerebral apparatus, proffering a substrate for the governance and impetus of human cognition and comportment. A robust executive function is a prerequisite for ensuring the progression of an individual’s mental well-being [49].

As children age and accumulate experience, their executive functions undergo rapid development. Inhibition, for instance, manifests as early as infancy, with its first significant leap occurring in the preschool years and continuing to improve throughout childhood [50]. Research indicated that infants could delay eating times, and the ability to delay eating increases with age. Specifically, 50% of two-year-olds could delay eating for 20 s, 85% of three-year-olds could inhibit the impulse to eat for one minute, and four-year-olds could delay eating for up to five minutes [51]. Luria’s tapping test revealed that children between the ages of four and four and a half showed marked improvements in inhibition, with most advances occurring before age six. Older children demonstrated faster response times and higher accuracy rates [52]. Concerning working memory, continuous improvements are observed from infancy through preschool. Perlman found that prefrontal cortex activation during a working memory task increased with age in children aged three to seven [53]. Best posited that individual working memory followed a linear trajectory from ages 4 to 14, stabilizing after 16 [50]. In a parallel vein, Ahmed utilized nationally representative data to exhibit nonlinear growth patterns in working memory performance from ages 3 to 19, with the most rapid growth occurring during childhood [54]. Research on cognitive flexibility suggests that this skill begins to emerge in children around the age of two and gradually develops between the ages of three and five [50]. Buttelmann et al.’s study supports this notion and indicates that cognitive flexibility develops rapidly in preschool and continues to increase into adolescence [55]. By age 12, children’s cognitive flexibility levels approximate those of adults [56].

The relation between video games and executive function is a complex and debated topic in psychology. Some studies argued that video games might have positive effects on specific aspects of executive function. For instance, Whitlock et al. discovered that engagement with the massively multiplayer role-playing game “World of Warcraft” enhanced Stroop performance [57]. Similarly, Liu et al. found that video gaming significantly elevated children’s inhibition abilities; children who had undergone video game training outperformed their non-trained counterparts on the Go/No-Go task [47]. Even if they no longer engage in gaming, individuals who were gamers before adolescence displayed superior working memory performance, heightened attentional focus, and enhanced information acquisition capabilities [58]. A meta-analysis by Glass et al. revealed that gaming conditions emphasizing the maintenance and rapid switching between multiple information and action sources substantially increased cognitive flexibility [59]. Further research indicates that complex puzzle-based video games, which necessitate strategic planning and reframing, can even augment the thickness of the player’s right dorsolateral prefrontal cortex, right hippocampal formation, and bilateral cerebellar cortex [60,61], thereby significantly elevating executive function levels.

Conversely, studies have elucidated that inappropriate exposure to video games could yield detrimental repercussions for children’s executive function [62]. Cognitive engagement in video gameplay is the mechanism of acute effects on executive function [63]. A study by Yang et al. unveiled an adverse correlation between the presence of action-oriented content in video games and a facet of executive function related to inhibition in children [64]. Correspondingly, scholarly exploration has indicated that participating in violent video games can modulate prefrontal cortical activity while engaging in cognitive inhibition [65]. Moreover, an escalating degree of addiction to video games corresponds to deteriorating performance in working memory tasks [66]. A systematic review offered empirical evidence that pathological and/or excessive utilization of video games engenders detrimental outcomes for cognitive processes, encompassing inhibition and decision-making [67]. Similarly, research showed that individuals exposed to video games for 3 h per day showed reduced inhibitory control compared with those exposed to video games for a limited amount of time per day. Those overexposed to video games had smaller gray matter volumes and thinner cortex in the ventral medial prefrontal cortex, along with shallower dorsolateral frontal sulci [68]. Hence, it can be logically inferred that excessive engagement with video games may harm children’s executive function, highlighting a critical area for further investigation and potential intervention.

Empirical evidence illustrates that heightened levels of executive function play a significant role in children’s social development [69]. A study by Hughes and Ensor found that executive function skills, such as inhibitory control and working memory, were positively associated with theory of mind (ToM) abilities, which involve understanding others’ thoughts and feelings [70]. Similarly, Ming et al. found that higher levels of executive function were associated with greater social competence in children [71]. In contrast, children with poorer executive function also experienced increased isolation and less engagement with peers on the playground [72]. Ego Depletion Theory posits that the performance of volitional activities—such as controlling processes, forming choices, initiating behaviors, and overcoming reactions—requires the expenditure of cognitive resources [73]. When a child’s executive function is debilitated, the tasks of self-control and regulation necessitate an atypical abundance of resources. A subsequent depletion of these resources may lead to an escalation in the child’s aggressive behavior. Empirical studies have further validated that executive function is a significant predictor of aggression levels in children, with children exhibiting deficits in this area being more inclined to demonstrate aggression in social interactions [74]. Given this evidence, it is plausible to hypothesize that engagement with video games is associated with children’s social development via the mechanism of executive function.

### 1.3. The Moderating Role of Age

Video game engagement exhibits a close relation with executive function, and investigating variables that may moderate the association between video game engagement and executive function holds significant value for enhancing children’s cognitive development. Among these moderating variables, age has been identified as an essential factor that may influence this relation [50]. Ecological Systems Theory emphasizes the importance of considering the systems and contexts that influence child development, including the role of age. Children’s age is a temporal system interacting with other ecological systems to shape their development [75]. Research has shown that children’s age relates to their video game engagement and its relation to executive function [76].

Recent evidence demonstrates a developmental trend in video game exposure, where older children engage in playing video games more frequently than their younger counterparts. A study illustrated that children’s exposure to video games was infrequent before age two, with average playtime recorded at approximately 20 min per day for children aged 2–3 years. This duration of exposure exhibited a progressive increase with age, with those between 5–8 years playing video games for an average of 40 min daily and those between 8–12 years for approximately 80 min a day [76]. Concordantly, Gentile found that older students were more prone to report excessive video game playing [77]. These convergent findings support the notion that older children engage more in video games than younger children, highlighting the importance of investigating potential implications for cognitive development and executive function.

In addition to the amount of time of video game exposure, children’s game genre preferences change throughout their lifespan. The categorization of video games varies across studies due to differing research objectives and target populations. Traditional classifications encompass various game types, from strategy and puzzle games to action/adventure and simulation games [78]. However, these categories often blur, leading researchers to develop customized typologies [79]. For example, Yu and Chan grouped games into four types based on player impact: conventional, exergames, cognitive training, and VR/simulation games [80]. Eichenbaum et al. tailored their classification to the needs of school-age children, identifying five game types: role-playing, action, strategy, music, and puzzles [81]. In the present study, a unique eight-type classification is adopted, specifically designed to align with the age characteristics of young children and the popularity of games. The categories are puzzle games, action games, simulation games, art games, sports games, adventure games, role-playing games, and other games. Video games are dedicated to meeting the needs of individuals, which change throughout the developmental stages as they grow in their abilities. For instance, the cognitive skills required for navigating game challenges are age-dependent; game genres that engage younger children may lose their appeal to adolescents [82]. Research has found that brain games that include exploration and decision-making elements are more popular with preschoolers [81,83]. Conversely, action-adventure games tend to captivate school-age children [84]. Some research has concluded that action games require players to focus on fast-paced, complex goals and that such a requirement may be too demanding for younger children [81].

Childhood is the optimal time to examine whether and how screen time exposure (such as video gaming) affects executive function development [50,82]. Assessments grounded in cognitive and neurophysiological methodologies reveal that executive function, though initially emerging during the foundational years of life, exhibits a marked trend of continual strengthening and maturation throughout childhood and adolescence [50]. The development of executive function is a dynamic process influenced by both biological maturation and environmental factors [50]. Video game engagement, particularly during the sensitive and formative stages of childhood, introduces a multifaceted environmental factor that can interact with the biological maturation of executive function. As children grow older, their likelihood of engaging with video games tends to increase, introducing a variable that may exert both positive and negative influences on the development of executive function. On the one hand, select video games can pose cognitive challenges that stimulate problem-solving capabilities, potentially contributing to executive function development [83]. Concurrently, specific games—especially those equipped with educational components or engineered for collaborative gameplay—have been associated with enhancements in cognitive performance and the cultivation of social skills [1,85]. On the other hand, excessive exposure to video games, especially those lacking educational value, may be detrimental, hindering the natural progression of executive function [67]. Consequently, a child’s age can function as a moderating variable within the connection between video game engagement and executive function, signifying that age could potentially intensify this specific association.

### 1.4. The Present Study

Literature on the association between video game engagement and children’s social development has predominantly focused on direct effects, leaving a critical gap in understanding the underlying mechanisms and moderating factors. Moreover, prior studies have yet to adequately explore how age may amplify the correlation between video game exposure and executive function or how video game engagement may deplete cognitive resources, subsequently affecting social development. Grounded in Ecological Systems Theory, which emphasizes the importance of a child’s age, and Ecological Techno-Microsystem Theory, which emphasizes electronic media’s role as a microsystem influencing development, the present study sought to fill this gap by constructing a moderated mediation model. The study specifically aims to (1) elucidate the direct correlation: rigorously examine the direct relation between the level of video game engagement and the level of social development in children; (2) clarify the mediating mechanism: evaluate executive function as a potential mediating variable in the linkage between video game engagement and social development in children; and (3) assess age-related moderation: investigate whether the child’s age moderates the effect of video game engagement on executive function, and, by extension, social development.

Based on the preceding contextual framework, the present study proposes the following hypotheses, visually articulated in the conceptual model shown in Figure 1: (1) An elevated level of video game engagement in children may be correlated with pronounced difficulties in social development, signifying a negative relation between children’s engagement with video games and their social growth; (2) executive function serves as a mediator in the nexus between video game engagement and social development, with an increase in video game engagement corresponding to a decrease in executive function, consequently culminating in diminished social development in children; and (3) children’s age could potentially moderate the connection between video game engagement and executive function, intensifying the association between the two.

## 2. Materials and Methods

### 2.1. Sample

The present study used convenience sampling, distributing anonymous questionnaires to teachers in various kindergarten and primary schools across Beijing. Teachers, in turn, extended the invitation to complete the questionnaires to parents residing with their children in Beijing. A crucial criterion for inclusion was that the children must have been exposed to video games. The teachers explicitly communicated this requirement during the distribution process. For families with multiple children, parents were instructed to concentrate their responses on a single child engaged in video gaming. The final sample of children comprised 431 participants, with 227 boys (52.7%) and 204 girls (47.3%), spanning ages from 3 to 9.7 years, with a mean age of 5.13 years (*SD* = 1.24). Of the children, 325 were only children (75.4%), and 106 children (24.6%) had siblings. A total of 313 children lived in cities (72.6%), 82 in towns (19%), and 36 in rural areas (8.4%). Concerning the age of first exposure to video games, 123 children were exposed before the age of 3 (28.5%), 304 between the ages of 3 and 6 (70.6%), and four after the age of 6 (0.9%).

### 2.2. Measures

#### 2.2.1. Video Game Engagement

The study employed the Video Game Engagement in Children Questionnaire, initially developed by Huang et al. and Qu et al. [8,86]. This questionnaire initially consisted of 20 items but was refined to 19 items across four dimensions after exploratory factor analysis by Qu et al. [8]. The dimensions encompass an interest in the activity, focus during play, challenges in discontinuation, and social disengagement. The questionnaire was scored on a 5-point Likert scale, and scores for the entire questionnaire ranged between 19 and 95 points. Higher total scores on the questionnaire indicate higher levels of engagement in video games. The Cronbach’s alpha coefficient of the overall questionnaire was 0.93. Furthermore, the questionnaire incorporated a multiple-choice query to discern the types of games frequently played by children. Options encompassed puzzle games (including card games and board games, among others), action games (including shooting games, fighting games, etc.), simulation games (including business management simulations and vehicle simulations, among others), art games (including music and artistic games, etc.), sports games (including exergames, among others), adventure games (including interactive movies, etc.), role-playing games (including MMORPGs, among others), and other types of games.

#### 2.2.2. Executive Function

The Children’s Executive Functioning Inventory (CHEXI), developed by Torrell, was used in this study [45]. The inventory has 24 items divided into three dimensions: inhibition, working memory, and cognitive flexibility [87]. The questionnaire incorporated questions that asked parents about their children’s behavioral performance in daily life scenarios (e.g., “Has difficulty refraining from smiling or laughing in situations where it is inappropriate”). Unlike assessments conducted in a structured laboratory setting, parents’ observations offer a unique and credible perspective, as they are based on children’s behavior across different scenarios [45]. Participants were instructed to respond to each item using a 5-point reverse scale because all the items in the questionnaire were phrased negatively. In this context, a higher score denoted a more pronounced disagreement with the statement, and a lower score signified more robust agreement. The inventory range of values is from 24 to 120 points. It was hypothesized that an increment in the total score would indicate better executive function development. Additionally, the reliability of the Children’s Executive Functioning Inventory was assessed, resulting in a Cronbach’s alpha coefficient value of 0.94, suggesting a high level of internal consistency.

#### 2.2.3. Social Development

The Children’s Social Development Scale (3rd edition), devised by Chen, was used to assess the social development of children [88]. This scale has been widely applied in China. It comprehensively evaluates children’s social development in various dimensions, including adherence to social rules, social cognition, volition, life habits, introversion and extroversion, attachment to family, emotional stability, self-concept, peer relationships, aggression, independence, honesty and fairness, empathy and helpfulness, competitiveness, and self-esteem. The scale consists of 60 items and is scored using a 5-point Likert scale. The total score on the scale spans from 60 to 300. A higher total score indicates better social development in children. In the present study, Cronbach’s alpha coefficient for the Social Development Scale was found to be 0.95, indicating good internal consistency.

### 2.3. Statistical Analysis

Before embarking on inferential analyses, the dataset underwent an initial cleaning process to address missing values, outliers, and inconsistencies. Preliminary analyses utilized descriptive statistics and Chi-square and MANOVA tests to understand the types of games most frequently played by children. The results of the analyses informed the inclusion of a control variable, action game involvement, which indicates whether a child frequently engages in action games.

The Jarque–Bera test for skewness and kurtosis was employed to evaluate each variable’s distributional properties. Skewness values ranged from −0.06 to 0.91, and kurtosis values from −0.07 to 1.25. These outcomes confirm the dataset’s adherence to the assumptions of normality, meeting the criteria for ensuing inferential analyses, as corroborated by Zhang [89].

The study targets primary variables: video game engagement, executive function, age, and social development. Descriptive statistics and Pearson correlation coefficients were calculated using IBM SPSS Statistics version 23.0. A moderated mediation analysis was conducted using PROCESS (version 4.0) software. This analysis employed a bootstrap method with 5000 resamples, exploring the mediating role of executive function and the moderating role of age in the relation between video game engagement and social development. The control variable, action game involvement, was integrated to account for its potential influence.

## 3. Results

### 3.1. Common Method Bias

Using a questionnaire to collect data in this study created a potential susceptibility to common method bias. Following the recommendations of Podsakoff et al. [90], we employed Harman’s single-factor test to investigate the presence of such bias. The findings revealed 25 factors with eigenvalues greater than 1. Moreover, the cumulative variance accounted for by the initial factor amounted to 19.02%, a value that fell below the critical threshold of 40% [91]. Hence, it can be inferred that this study is not affected by any significant common method-bias issue.

### 3.2. Preliminary Analysis

A descriptive analysis concerning the most played types of video games among children revealed an average engagement in 2.71 distinct game categories (*SD* = 1.31, *Minimum* = 1, *Maximum* = 8). A Chi-square analysis substantiated that puzzle games were most prevalent among children, succeeded by action, simulation, art, role-playing, and other game genres. Comparatively, adventure and sports games had a less frequent engagement, χ^2^ = 642.11, *p* < 0.001.

Further statistical scrutiny employing MANOVA tests (refer to Table 1) disclosed that children who habitually engage in action games manifested elevated levels of video game engagement (action gamer: *M* = 56.94, *SD* = 13.93; non-action gamer: *M* = 52.76, *SD* = 12.07) and diminished executive function scores (action gamer: *M* = 70.48, *SD* = 14.69; non-action gamer: *M* = 74.70, *SD* = 14.06) compared to their counterparts who rarely indulge in such activities. Age-wise, children who recurrently partake in puzzle games tend to be younger (*M* = 5.07 years, *SD* = 1.19) than those who do not (*M* = 5.45 years, *SD* = 1.40), while the converse holds for action games (action gamer: *M* = 5.30 years, *SD* = 1.27; non-action gamer: *M* = 5.04 years, *SD* = 1.21). No observable disparities were discerned for children who frequently or infrequently engage in other gaming categories regarding their levels of video game engagement, executive function, social development, or age. Consequently, to control for the possible confounding effects of game types on the interrelationships between video game engagement, executive function, social development, and age, whether individuals frequently engage in action games (named action game involvement) was incorporated as a control variable in ensuing model-based analyses.

### 3.3. Descriptive Statistics and Correlation Analysis

Table 2 presents the means, standard deviations, and correlation analysis results for the primary variables examined in this research. Children’s engagement with video games exhibited a negative relation with both their executive function and social development. Conversely, a positive correlation was detected between executive function and social development. Furthermore, children’s age positively correlated with their social development.

### 3.4. Moderated Mediation Effect Test

PROCESS version 4.0, a statistical tool developed by Hayes [92], was used to examine the mediating role of executive function and the moderating role of children’s age in the relation between engagement with video games and social development. A Bootstrap test for confidence intervals was performed on the model to correct for bias, with a replicated sample of 5000 and a confidence interval of 95%.

#### 3.4.1. The Mediating Effect of Executive Function

Based on the findings in Table 2, statistically significant direct relations emerged between video game engagement, executive function, and social development. These outcomes fulfilled the prerequisites for undertaking a mediation analysis. Given that the preliminary analysis suggests a correlation between action game involvement and increased levels of video game engagement in children, which may be associated with diminished executive function, we deemed it crucial to incorporate action game involvement as a control variable. These findings collectively laid the foundation for proceeding with a mediation analysis. To examine the potential relation between children’s engagement in video games and their social development through executive function, we opted for Model 4 out of the 76 conventional models put forth by Hayes to conduct mediation effects analysis [92].

The results delineated in Table 3 indicated several significant associations. Firstly, a discernible negative relation was observed between action game involvement and social development (β = −0.10, *t* = −2.13, *p* < 0.05). Secondly, there was a notable negative association between video game engagement and executive function (β = −0.65, *t* = −17.50, *p* < 0.001). Lastly, a positive correlation was detected between executive function and social development (β = 0.28, *t* = 4.61, *p* < 0.001).

In contrast, upon introducing executive function as a mediating variable, the initial correlation between video game engagement and social development (β = −0.16, *t* = −3.42, *p* < 0.001) no longer retained its significance (β = 0.02, *t* = 0.31, *p* > 0.05). The findings indicated a significant mediation effect (*ab* = −0.35, 95% CI [−0.53, −0.19]), with executive function acting as a complete mediator between children’s video game engagement and social development (Table 4). In other words, children who engage in video games more often are more likely to have impaired executive function, which is detrimental to their social development.

#### 3.4.2. The Moderating Role of Children’s Age

Model 7, developed by Hayes [92], was employed to examine the presence of a moderated mediation effect (Table 5). The mediation model introduced children’s age as a moderator variable, and action game involvement remains the controlling variable. The interaction term was associated with executive function, exhibiting a negative coefficient and, thus, a substantial negative relation (β = −0.11, *t* = −3.08, *p* < 0.01). The finding indicated that age moderates the relation between children’s video game engagement and executive function. The moderated mediation model is shown in Figure 2.

We also conducted a simple slope analysis to illustrate the role of the interaction term (Figure 3). In the context of younger children, video game engagement was negatively associated with executive function (β = −0.55, *p* < 0.001, 95% CI [−0.65, −0.45]). However, this negative association between video game engagement and executive function became more pronounced in older children (β = −0.76, *p* < 0.001, 95% CI [−0.86, −0.66]). Importantly, Figure 3 and its associated interpretations should not be taken as a comparison of absolute levels of executive function between younger and older children. Instead, the emphasis is on how the strength of the negative correlation between video game engagement and executive function varies with age, becoming increasingly pronounced as children grow older.

## 4. Discussion

In today’s media-saturated environment, video games have become a fundamental component of children’s daily experiences. However, scholarly perspectives on the implications of video games for children’s development remain diverse [5]. A cohort of researchers holds a favorable opinion towards video games, arguing that video games provide opportunities for children’s cognitive [93,94,95,96], social [97,98] and emotional [99,100] development. Conversely, other scholars have underscored potential detriments, suggesting that video games can adversely influence children’s physical health [101,102], academic performance [103,104], and social skills [28,105]. A comprehensive scientometric and knowledge-mapping analysis regarding the impact of video games on child development revealed that contemporary research recognizes video games as a double-edged sword concerning child development [5]. Specifically, this research has elucidated that excessive engagement in video games may harm various dimensions of child development.

### 4.1. Video Game Engagement and Social Development

This study revealed a negative correlation between video game engagement and social development. Notably, children who engage more in video games exhibit poorer social development. This finding is consistent with Hypothesis 1 of the research. In a systematic review, the authors noted that people engage in video games for several reasons [106]. At the social level, game engagement is associated with relationships. Players are motivated by the need to connect and be approved by friends, and social interaction impacts game engagement [106]. However, as engagement with video games intensifies, the close and enduring connections formed between participants within the gaming environment may create a displacement effect. Such an effect could reduce the quantity and quality of offline communication, resulting in a diminished social circle and difficulties in developing or sustaining social and emotional skills [107,108]. Prior research has consistently observed that children’s interest and focus on video games often lead to challenges in disengaging from the games. Such deep engagement with video games diverts time and energy from real-world social interactions, subsequently negatively influencing social development [109].

According to the Reduced Social Cues Theory, the diminution of social cues within online communities can weaken social norms and constraints, giving rise to an online disinhibition effect [110,111]. This phenomenon may cause individuals to express themselves in ways they typically avoid in face-to-face interactions, including heightened aggression and abnormal behaviors [112]. Moreover, prolonged exposure to video games may increase the risk of internet addiction among children, leading to greater investment in gaming and subsequent social avoidance. This misalignment with real-world interactions can result in diminished socialization and well-being [113]. The influence of violent video games may further compound this issue, contributing to increased aggression in children. Research has demonstrated that even short-term exposure to violent content can elevate perceptions and behaviors related to aggression [105]. Cumulatively, children’s engagement with video games can manifest in detrimental ways, affecting emotional states, social cognition, aggression, and self-esteem and significantly impeding their social development.

The COVID-19 pandemic has significantly complicated these dynamics, particularly in China, where stringent social restrictions have existed for extended periods. Measures like school closures and home quarantines have curtailed children’s opportunities for physical activities and interpersonal interactions. This has led to increased video game engagement, with studies by Alsaad et al. and Mohan et al. showing a significant rise in the duration of children’s video game activities during quarantine [114,115]. Notably, this increase varied by age; children older than 12 months showed a rise in media usage, whereas those younger than 12 months remained relatively stable [116,117].

The pandemic’s influence has also been profound on the social development front. Home quarantine has led to various psychological issues in children, including reduced social interactions and emotional well-being [118]. Wise’s research revealed that infants born during the pandemic exhibit weaker social skills than those born before [119]. Similarly, a study by Deoni et al. suggests a decline in cognitive abilities among children born during the pandemic relative to those born prior to its onset [120]. The researchers speculate that this decline may be due to the ubiquitous wearing of masks in public places, schools, and daycare centers, which could potentially impact the development of attachment and social-emotional processing skills, thereby affecting their social and emotional development. Furthermore, Teng et al. found that social restrictions increased levels of video game disorders in children, negatively impacting their social and emotional well-being [121]. In line with this, Alsaad et al. reported that 20.4% of children became more introverted, and 14.5% became more aggressive after playing video games during social restrictions [114].

Our study also found that action games significantly increase children’s engagement in video games and negatively predict their social development. This finding is supported by other studies indicating that action games can be particularly engrossing due to their rich visual and auditory stimuli, unpredictable actions, and immediate reward mechanisms. However, they can also reduce players’ empathy and increase aggressive behavior [122]. Long-term exposure to action games may lead to negative emotional states in slower-paced, less stimulating social environments [123].

### 4.2. The Mediating Role of Executive Function

Our findings corroborate that executive function serves as a full mediator in the relation between video game engagement and social development, thereby substantiating Hypothesis 2. Specifically, the study identified a negative correlation between reported video game engagement and the level of executive function, revealing that excessive involvement in video games may be associated with impairment in executive function. This impairment in executive function, in turn, is linked to detrimental social development. These insights underscore the importance of measured playtime and diligent supervision, all essential in fostering a child’s balanced and healthy development of executive function and socialization.

Video game engagement was negatively associated with children’s executive function. Grounded in Reinforcement Sensitivity Theory, this phenomenon can be understood through the behavioral activation system, which orchestrates an individual’s behavior in response to rewards, often leading to increased behavioral persistence and impulsivity [124]. Playing video games is perceived as a rewarding behavior, and the brain’s intrinsic reward mechanism fuels children’s engagement with video games [125,126,127]. However, when this engagement becomes excessive, it may diminish an individual’s inhibition. High levels of play engagement can lead to extended playtime and diminished connectivity within the frontal regions of the brain, resulting in impaired behavioral inhibition [105,128,129]. This perspective aligns with findings from studies on individuals with Internet gaming disorder [126,130], further underscoring the complexity of the relation between video game engagement and executive function. Additionally, children who frequently engage in action games exhibit lower scores in executive function. This observation is partly explained by research from West et al., who found that habitual engagement in action games is associated with reduced gray matter in the hippocampus, a crucial brain area for working memory cognitive functions [131].

The connection between executive function and social development lends credence to the Self-Depletion Theory. This theory posits that when executive function becomes impaired, the individual expends remaining cognitive resources to self-regulate and repair, consequently depleting the reserves that would otherwise be available for social development. Neurological evidence indicates that the prefrontal lobe plays a pivotal role in constraining the level of executive function and individual social development sub-functions, such as emotion regulation, social interaction, and social cognition [132,133,134]. Therefore, any impairment in executive function is likely to have cascading effects on social development. Conversely, a well-developed executive function furnishes the essential cognitive resources required to participate in intricate social interactions and comprehend others’ mental states [70]. In a parallel vein, recent research has corroborated that executive function is indispensable in facilitating adaptive social behavior and meaningful social interactions [135]. These insights serve as a valuable guide for parents, emphasizing the necessity of monitoring and controlling the intensity of their children’s engagement with video games. Parents should safeguard their children’s overall cognitive and social well-being by taking appropriate measures to avoid children’s excessive or prolonged exposure to video games.

### 4.3. The Moderating Effects of Age

A notable finding from the study is the age-moderated relation between video game engagement and executive function. With increasing age, the negative correlation between engagement in video games and the development of executive function became more pronounced, paralleling the intensification of the adverse outcomes associated with impaired executive function on children’s social development. These observations support Hypothesis 3, shedding light on the nuanced interplay between age, video game engagement, and cognitive and social growth. Additionally, our study discovered that younger children prefer puzzle games, while older children are more inclined towards action games. This preference aligns with their developmental stages. Puzzle games, characterized by a slower pace and fewer cognitive demands, are more easily comprehensible and adaptable for younger children. In contrast, school-aged children are drawn to the stimulation that action games provide, often viewing victory or defeat as the sole metric for performance assessment [136]. These games, commonly set in fictional and fantastical worlds, cater to their heightened curiosity [137]. For older children, the preference for action games was reported for their challenging nature and as a means to relax and escape life’s worries [137].

Dynamic Systems Theory posits that the development of children’s executive function is the product of the interaction between children and their environment [138,139]. Unfavorable environmental conditions during childhood can pose significant risks to future psychological development [140]. Therefore, an increasing number of studies have begun to focus on the role of age in the relation between media use and adverse cognitive development [141]. The age-related intensification of the negative correlation between video gaming and executive function emerges as a complex phenomenon shaped by developmental and neurological considerations. In the period leading up to adolescence, children’s engagement with video games tends to increase [76,142], and this pattern is mirrored by a rise in gaming addiction that follows an inverted U-shaped trajectory, reaching a peak during adolescence [143]. This progressive escalation in exposure to video games strengthens the negative association with executive function, creating a more pronounced effect as age advances.

Adding to this complexity is the role of the frontal lobe, a region widely recognized for its crucial contribution to executive function [144]. Long-term exposure to electronic media can impede the typical development of the frontal lobe [63,145]. The neural developmental changes that occur during childhood and adolescence may foster an imbalance between the brain’s emotional centers and the executive control mechanisms housed in the frontal lobe [146,147]. This imbalance becomes more prominent as children grow older and can be further exacerbated by excessive engagement with video games.

The study may serve to guide parents and educators in promoting healthy media consumption habits. For example, findings highlight the need to select age-appropriate game content and understand the nuanced interplay between age and video game engagement. This age-related intensification also calls for targeted intervention strategies. Tailoring interventions based on age may lead to the more effective prevention and treatment of gaming addiction and related cognitive impairments.

### 4.4. Limitations and Suggestions for Future Research

The current research, focused on the connections between video game engagement and social development in children, presents valuable insights but also has limitations that necessitate careful consideration. One such area for improvement is the study’s confined sample size. Although participants were drawn from several urbanization levels within Beijing, the generalizability of the findings to other populations may be limited. Future studies should include a broader and more nationally representative sample to overcome this constraint, thereby enhancing the applicability of the results across different sociocultural contexts.

A significant challenge in the current study lies in the absence of a causal experimental or longitudinal follow-up design, hampering the identification of causal associations among the variables. This limitation underscores the importance of adopting experimental or longitudinal methodologies in subsequent investigations, which would provide more concrete evidence of causality and strengthen the validity of the findings.

Furthermore, our Video Game Engagement Questionnaire should be enhanced to meticulously quantify the frequency and duration of children’s interactions with various video game genres and content. Prior research has shown that various game categories, such as pro-social and cooperative games, positively affect children’s executive function and social development [148,149]. In contrast, our data reveals a negative association between action game involvement and both executive function and social development. Thus, it becomes pivotal for ensuing research to further hone these measurement tools, ensuring an in-depth grasp of the multifaceted influences of video game types on children’s developmental trajectories.

In addition to these limitations, other variables could significantly influence the relationship between video game engagement and social development. Ecological factors, such as parenting styles and socioeconomic status, have substantially impacted children’s social interactions and video game play [86,150]. Likewise, mental health status and life stress levels interact with video game exposure, potentially offering trauma-repair experiences [151,152]. Thus, future research should aim to incorporate these additional variables as control or moderating factors. This multidimensional approach will enhance the robustness and generalizability of future findings and contribute to a more nuanced understanding of the complex relation between video game engagement and children’s social development.

### 4.5. Implications

Understanding the complex interplay between children’s engagement with video games, executive function, age-related factors, and social development is critically important. This study uncovers a negative correlation between unregulated video game exposure and executive function, a critical cognitive precursor for social development. Further complexity is introduced by the moderating role of age, indicating the necessity of stage-specific interventions.

In a digital milieu, it is impractical to place ‘digital natives’ in a ‘de-electronic gaming vacuum’, given that a negligible fraction of children aged 1–13 have never engaged with electronic media [149,153]. Hence, the role of parents in controlling screen time becomes particularly vital, especially in light of evidence that children are dedicating up to 6 h daily to technology [154,155]. Effective parental regulation transcends mere monitoring of the environment and playtime and requires adhering to age-appropriate media consumption guidelines. For example, the American Academy of Pediatrics advocates limiting high-quality media exposure for children aged 2–5 to one hour daily while encouraging parental co-viewing and guidance [156].

As video games have become an integral part of the daily lives of children who are ‘digital natives,’ parents and educators must consider their role in shaping children’s gaming habits [1]. The prudent selection of age-appropriate content constitutes a pivotal aspect of this responsibility. This selection should align with guidelines from reputable organizations such as the Entertainment Software Rating Board (ESRB) while also considering insights into the suitability of games for different age groups, which can be obtained from mobile download platforms like the Apple Store [1]. In this context, the focus should extend beyond merely limiting exposure to action games to a broader strategy aimed at harnessing the potential of video games for positive developmental outcomes.

In conclusion, the intricate factors affecting children’s video game engagement require a collaborative, multi-sectoral approach involving educators, parents, and policymakers. Regulatory efforts, such as China’s recent limitation on online gaming for minors and the implementation of real-name registration systems, are examples of regulatory strategies that can be deployed.

## 5. Conclusions

The study illuminated a significant and negative relation between children’s engagement with video games and their social development. Executive function emerged as a complete mediator in this relation, elucidating the underlying mechanisms connecting video game engagement and social growth. Additionally, the study discerned a moderating role of children’s age on the relation between video game engagement and executive function. Namely, the correlation between video game engagement and executive function intensified with maturation, reflecting a more pronounced connection with age.

## Figures and Tables

**Figure 1 behavsci-13-00833-f001:**
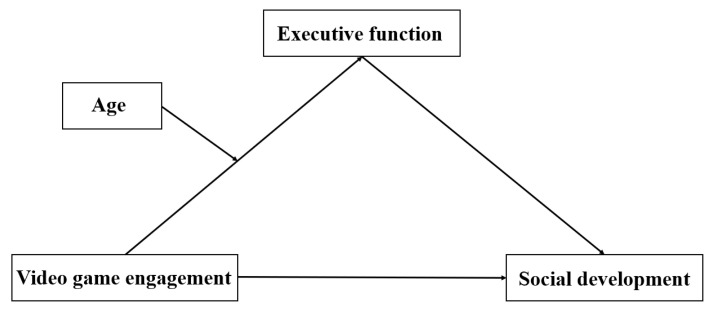
The moderated mediation model.

**Figure 2 behavsci-13-00833-f002:**
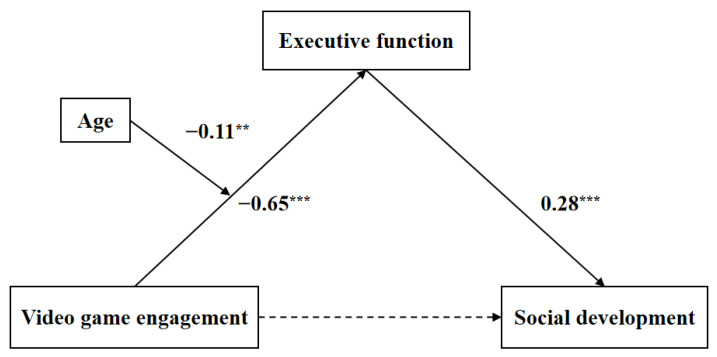
A moderated mediation model with standardized regression coefficients. ** *p* < 0.01, *** *p* < 0.001.

**Figure 3 behavsci-13-00833-f003:**
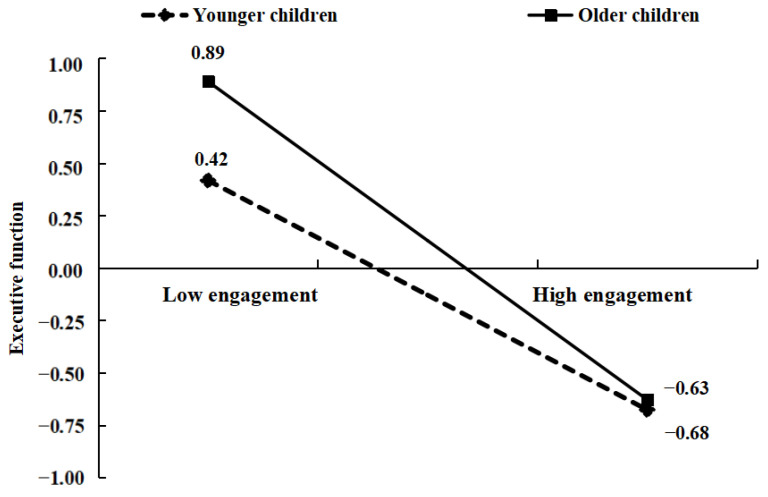
A simple slope for interaction between video game engagement and children’s age in executive function.

**Table 1 behavsci-13-00833-t001:** MANOVA analyses evaluating disparities in key variables based on children’s inclination to play specific game genres often.

Game Genre	Response		Percent of Cases (*n* = 431)	*F* Value			
	*n*	Percent		VGE	EF	SD	Age
Puzzle	362	31.05%	83.99%	2.08	0.33	0.52	5.64 *
Action	144	12.35%	33.41%	10.32 ***	8.40 **	2.58	4.02 *
Simulation	139	11.92%	32.25%	0.18	0.74	1.86	0.86
Art	134	11.49%	31.09%	0.22	1.79	3.02	1.11
Sports	39	3.34%	9.05%	0.94	2.08	0.93	1.46
Adventure	91	7.80%	21.11%	1.84	0.75	0.03	0.63
Role-playing	129	11.06%	29.93%	1.323	1.75	0.22	0.08
Other	128	10.98%	29.70%	1.66	3.39	0.00	0.64
Total	1166	100.00%	270.53%				

*Notes*: VGE = video game engagement, EF = executive function, SD = social development. * *p* < 0.05, ** *p* < 0.01, *** *p* < 0.001.

**Table 2 behavsci-13-00833-t002:** Descriptive statistics and correlations of the main variables.

Variables	*M*	*SD*	1	2	3	4
1. Video game engagement	54.16	12.86	1			
2. Executive function	73.29	14.40	−0.65 **	1		
3. Social development	218.82	24.74	−0.15 **	0.26 **	1	
4. Age	5.13	1.24	0.09	0.05	0.11 *	1

*Notes*: * *p* < 0.05, ** *p* < 0.01.

**Table 3 behavsci-13-00833-t003:** Test of the mediating effect of executive function.

Result Variable	Predictor Variable	*R*	*R^2^*	*F* Value	β	*t* Value
Social development		0.18	0.03	7.17 ***		
	Action game involvement				−0.10	−2.13 *
	Video game engagement				−0.16	−3.42 ***
Executive function		0.65	0.43	160.35 ***		
	Action game involvement				0.04	1.06
	Video game engagement				−0.65	−17.50 ***
Social development		0.28	0.08	12.09 ***		
	Action game involvement				−0.11	−2.41 *
	Video game engagement				0.02	0.31
	Executive function				0.28	4.61 ***

*Notes*: The variables in the model had been standardized and substituted into the equation. * *p* < 0.05, *** *p* < 0.001.

**Table 4 behavsci-13-00833-t004:** Analysis of all effects in a mediated model.

Variables	Effect Size	Boot SE	95% CI
Total effect	−0.32	0.09	[−0.50, −0.13]
Direct effect	0.04	0.12	[−0.20, 0.27]
Mediating effect of executive function	−0.35	0.09	[−0.53, −0.19]

**Table 5 behavsci-13-00833-t005:** Test of the moderating effect of children’s age.

Result Variable	Predictor Variable	*R*	*R^2^*	*F* Value	β	*t* Value
Executive function		0.67	0.45	88.33 ***		
	Action game involvement				0.05	1.27
	Video game engagement				−0.65	−17.96 ***
	Age				0.13	3.57 ***
	Video game engagement × Age				−0.11	−3.08 **

*Notes*: The variables in the model had been standardized and substituted into the equation. ** *p* < 0.01, *** *p* < 0.001.

## Data Availability

The dataset of this study is obtainable from the corresponding author, subject to reasonable requests.

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
