# Peer review of "Relations between Video Game Engagement and Social Development in Children: The Mediating Role of Executive Function and Age-Related Moderation"

_behavsci, 2023, doi:10.3390/bs13100833_

Round 1

Reviewer 1 Report

I appreciate the research effort presented in the paper "Relationship between Video Game Engagement and Social Development in Children: The Mediating Role of Executive Function and Age-Related Moderation." The authors address a highly relevant topic concerning the relationship between video game engagement and its potential impact on children's social development. 

While the study's findings contribute to the discourse, I believe it could be beneficial to introduce an alternative view of the results, taking into consideration the diverse perspectives on the role of video games in children's lives. A balanced approach would involve discussing both the potential negative aspects and the positive contributions that video games might offer to children's development.

Numerous studies have highlighted the potential negative effects of excessive video game engagement on social development, aligning with the negative relationship reported in your study. However, it is also worth noting that recent research has indicated that video games can have positive impacts on cognitive skills, problem-solving abilities, and even social interactions. Certain video games, especially those designed with educational elements or collaborative gameplay, have been associated with enhanced cognitive function and social skills. 

Introducing findings from studies that highlight the positive aspects of video game engagement could provide a more comprehensive view of the topic. For example, research has shown that some video games can improve decision-making skills, spatial awareness, and teamwork. By acknowledging these positive attributes, your paper would offer a more balanced perspective on the impact of video games on children's social development, i.e.: Raventós, C. L., & Belli, S. (2019). Psychological Outlook on Video Games: an Example of the Long Road to the Invention of Disorders not Associated with Substances.

In conclusion, while the negative relationship between video game engagement and children's social development is a crucial aspect to consider, incorporating an alternative view that emphasizes positive aspects could enrich the discussion. By acknowledging that video games are not uniformly detrimental to development, you can provide readers with a more nuanced understanding of the topic. This approach would further contribute to the ongoing conversation about the complex interactions between video games and children's development.

Minor editing of English language required

Reviewer 2 Report

The theme that the authors approach in the study is very relevant, especially for the post-pandemic society, which needs to deal now with the effects of using technologies for an extended period, as a substitute for face-to-face interactions. 

Introduction. The authors provide a literature review and a well-grounded conceptualization of video games, as well as some existing studies that focus on the concepts the paper is dealing with. The concept of video game engagement is detailed, with consistent references to various factors that influence children's developmental process. Both positive and negative elements of this impact are described. However, the question raised by point 1.1. in the article is the following: "Is there a difference in terms of the impact of video gaming before and after the COVID-19 pandemic?". For small children, especially the ones born during the pandemic, or that were very little at that time, the disruption caused by the restrictions imposed by governments all over the world left little space for authentic social relationships outside their own families. This, cumulated with the typical elements of the development of social interaction patterns at that age, might have created a particular framework for the use of technologies and video games, that the authors also need to address.

Section 1.2. describes the executive function, based on a literature review. The concept is presented in terms of facets, the relationship with video games, and the relationship with children's social development. However, there is a need for an extended discussion about how social competence typically looks in children at a particular stage of development. What can we expect from a 4-year-old child, a 5-year-old child, and so on?

Section 1.3. treats the moderating role of age in the relationship between video games and executive function. Is the paper considering a specific type of video game? Although the principles of video gaming are the same for every type of game, the fact that these can be classified into different categories might have different consequences. For example, puzzles might differ in their impact from FPS (First-Person-Shooter) games. This is an aspect that the paper should take into consideration.

The authors detail the sample, the variables they measured, and the statistical analyses they made, according to the appropriate methodological endeavor. The instruments used for the measurements are clearly described and cited. The hypotheses are clearly stated, and the statistics are supported by relevant tables and figures. 

The discussion section. Based on the age of the children in the sample, the discussion section should also address extensively the role of social restrictions these children had to face from a very young age, particularly since China has been one of the countries in which these restrictions were kept for a very long time, while in other parts of the world, governments had different approaches about dealing with the pandemic. There might be a difference between children born during the pandemic and children who were 3-4 years old at that time in terms of social development and video game consumption. Furthermore, there are studies that suggest the fact that for today's generations of children, the use of technologies (including video games) is normal for up to 6 hours a day, provided parents control the contents of this activity. Not allowing them or restricting them might in fact create difficulties in adjusting to the specifics of the life that this era holds. 

The limits and implications of the study are also debated, and the authors state the type of games was not a factor they took into consideration. However, there is a strong need to do so because otherwise, results might have difficulties in being generalized.

Reviewer 3 Report

The theoretical part is well structured.

The objectives need restructuring and concretisation.

Sample - the method of selection of research participants is not presented.

Statistical analysis - must show concretely the variables and methods used. Also, is the distribution of the data symmetric or asymmetric?

The results are presented scientifically.

Discussions are in connection with relevant literature.

You will need to help the authors correct potential errors.

Reviewer 4 Report

This paper investigated the detrimental effect of video game engagement on social development in children. Specifically, the authors focused on the mediating role of executive function and age-related moderation in the process of the effect. A sufficient review, discussion, and analysis highlight the high quality of this paper. The results were interesting. The following are some comments for improvement.

1. Relationship indicates a psychological bond in a specific interpersonal relation in developmental psychology. Therefore, I ask the authors to use “relation” instead of “relationship”.

2.  Parents evaluated their children’s video game engagement, executive function, and social development. While I agree with the procedure that the parents assessed their children’s engagement in games and social development, assessing executive function by parents is questionable since it is an internal and cognitive ability. Thus, I ask the authors to show a rationale for this. Is it behavioral level assessment and observable?

3.  Figure 3 shows that video game engagement potentially surpasses or spoils age-related growth in executive function. The description that the negative association between video game engagement and executive function became more pronounced as children grew older is correct. However, for me, it might give potential readers a wrong interpretation. If the score of executive function in older children were lower than in younger children on the case in high engagement for video games, the authors’ description would have been understandable.

4. Please indicate other potential variables to associate with social development in the limitation section. They might be control variables on the regression model.

The papre invloves good qualtiy of English.

Round 2

Reviewer 1 Report

I suggest to publish this manuscript in this journal.